# Rodent Models of Huntington’s Disease: An Overview

**DOI:** 10.3390/biomedicines11123331

**Published:** 2023-12-16

**Authors:** Giulio Nittari, Proshanta Roy, Ilenia Martinelli, Vincenzo Bellitto, Daniele Tomassoni, Enea Traini, Seyed Khosrow Tayebati, Francesco Amenta

**Affiliations:** 1School of Medicinal and Health Products Sciences, University of Camerino, Via Madonna Delle Carceri, 9, 62032 Camerino, Italy; giulio.nittari@unicam.it (G.N.); proshanta.roy@unicam.it (P.R.); ilenia.martinelli@unicam.it (I.M.); vincenzo.bellitto@unicam.it (V.B.); enea.traini@unicam.it (E.T.); khosrow.tayebati@unicam.it (S.K.T.); 2School of Biosciences and Veterinary Medicine, University of Camerino, Via Gentile III da Varano, 62032 Camerino, Italy; daniele.tomassoni@unicam.it

**Keywords:** Huntington’s disease, animal models, nervous system, rodents, genetics

## Abstract

Huntington’s disease (HD) is an autosomal-dominant inherited neurological disorder caused by a genetic mutation in the IT15 gene. This neurodegenerative disorder is caused by a polyglutamine repeat expansion mutation in the widely expressed huntingtin (HTT) protein. HD is characterized by the degeneration of basal ganglia neurons and progressive cell death in intrinsic neurons of the striatum, accompanied by dementia and involuntary abnormal choreiform movements. Animal models have been extensively studied and have proven to be extremely valuable for therapeutic target evaluations. They reveal the hallmark of the age-dependent formation of aggregates or inclusions consisting of misfolded proteins. Animal models of HD have provided a therapeutic strategy to treat HD by suppressing mutant HTT (mHTT). Transgenic animal models have significantly increased our understanding of the molecular processes and pathophysiological mechanisms underlying the HD behavioral phenotype. Since effective therapies to cure or interrupt the course of the disease are not yet available, clinical research will have to make use of reliable animal models. This paper reviews the main studies of rodents as HD animal models, highlighting the neurological and behavioral differences between them. The choice of an animal model depends on the specific aspect of the disease to be investigated. Toxin-based models can still be useful, but most experimental hypotheses depend on success in a genetic model, whose choice is determined by the experimental question. There are many animal models showing similar HD symptoms or pathologies. They include chemical-induced HDs and genetic HDs, where cell-free and cell culture, lower organisms (such as yeast, *Drosophila*, *C. elegans*, zebrafish), rodents (mice, rats), and non-human primates are involved. These models provide accessible systems to study molecular pathogenesis and test potential treatments. For developing more effective pharmacological treatments, better animal models must be available and used to evaluate the efficacy of drugs.

## 1. Introduction

Huntington’s disease (HD) is a neurodegenerative disorder in which, like any other neurodegenerative disease, a monogenic, completely penetrant protein misfolding occurs. HD is caused by a gene called huntingtin (HTT) [1,2], which was discovered 17 years ago. After this discovery, much information was obtained about the etiology of the disease. This is a progressive neurological disorder with a specific phenotype that includes chorea and dystonia, incoordination, cognitive impairment, and behavioral difficulties. In general, symptoms appear in middle life after afflicted people have children. However, HD can appear at any age between infancy and senescence [3]. Long-term memory is spared in HD, but executive functions, such as organizing, planning, checking, or adapting to alternatives, are impaired, and the acquisition of new motor abilities is delayed. These characteristics diminish over time, and speaking deteriorates faster than comprehension. Psychiatric and behavioral symptoms, unlike cognition, occur frequently [4,5,6,7]. HD causes notably selective neuropathological alterations, with significant cell loss and atrophy in the putamen and caudate [8,9]. HD primarily affects the striatum and cerebral cortex, as evidenced by extensive inclusion development and cell death in the striatum’s medium-sized spiny neurons (MSNs), which produce gamma-aminobutyric acid (GABA), and the cortex’s large glutamatergic pyramidal neurons [10,11]. MSNs in the striatum are the most vulnerable. Those that include enkephalin, the exterior globus, and neurons in the pallidum are more involved than cerebrocortical neurons projecting substance P to the interior globus [8,9]. Interneurons are unaffected. Several pieces of data support the theory that chorea predominates in the early stages of HD due to the indirect channel of the basal ganglia-thalamocortical circuitry’s preferred involvement [12]. The substantia nigra, cortical layers 3, 5, and 6, the CA1 region of the hippocampus [13], the angular gyrus in the parietal lobe [14,15], Purkinje cells of the cerebellum, the lateral tuberal nuclei of the hypothalamus [16,17], and the centromedian-parafascicular [18] complex of the thalamus are all affected by HD. HD pathogenesis is relevant to identifying prospective treatment targets. HTT is a big protein made up primarily of HEAT repeats, which are repeat units of roughly 50 amino acids. These repeats are made up of two antiparallel helices with a helical hairpin arrangement that combine to form a super helical structure with a continuous hydrophobic core [19]. HTT’s biological functions have not yet been clarified [3,20]. The protein is predominantly cytoplasmic, with palmitoylation at cysteine 214 [21], allowing it to adhere to the membrane. Near the C-terminus, there is a possible nuclear export signal, but no unambiguous nuclear localization signal has been identified. HTT is a nucleoprotein that shuttles into the nucleus, participates in vesicle transport, and regulates gene transcription [22]. It may play a role in RNA trafficking [23]. This paper reviews the main in vivo rodent models of this rare disease. A document search was conducted using the available literature extracted from PubMed (Medline), Cumulative Index to Nursing and Allied Health Literature (CINAHL), Google Scholar, Cochrane Library, and Biomed Central, applying Medical Subject Headings (MeSH). The search keywords were Huntington’s disease plus in vivo models or diagnosis or treatment. The analysis included papers published in the ten years from 1 January 2012 to 31 December 2022, as well as the articles cited in the selected papers. The inclusion criteria were that the title or abstract should contain at least one keyword corresponding to the search terms besides HD. All the articles considered were published in reputed peer-reviewed journals.

## 2. Utility and Validity of Animal Models of Huntington’s Disease

The ideal animal model of a given disease should reproduce the full spectrum of features with a time course like how the disease evolves in patients. Because disease features are characterized by pathological events on common neurobiological substrates, treatments improving key features of a pathology may improve broader symptoms, including those difficult to model. Several rodent HD models have been developed since the HTT mutation was identified as the cause of HD (Figure 1). Due to the enlarged polyglutamine stretch in mutant HTT, the protein acquires a toxic function, leading to functional impairment. The most widely studied invertebrate animal models, like *C. elegans* and *Drosophila melanogaster*, allow for rapid testing of specific hypotheses and novel therapeutic strategies. Expanded polyglutamine repeats are expressed in the brain of the worm in the *C. elegans* model [24]. There are two categories of animal models for HD: non-genetic models and genetic models. The main data obtained in these modelling studies using rodents are detailed in this paper.

## 3. Non-genetic Models of Huntington’s Disease

Nongenetic models are commonly used for excitotoxic processes or mitochondrial impairment, which induce cell death. Quinolinic acid (QA) and kainic acid (KA) were the most commonly excitotoxic agents used in both rats and primates to develop HD models. Excessive activation of glutamate-gated N-methyl D-aspartate receptors (NMDARs) due to increased glutamate exposure causes excitotoxicity, which leads to neuronal damage and death. Mitochondrial energy failure results from excessive Ca^2+^ release [25]. In striatal neurons, QA and KA are bound to their cognate receptors, N-methyl-D-aspartic acid (NMDA) and non-NMDA, inducing cell death. The mitochondrial toxins are 3-nitro propionic acid (3-NP) and malonic acid (MA) and were used to generate a rodent metabolic model [26].

### 3.1. Excitotoxin Models

KA produces remote lesions and, at high concentrations, destroys fibers on passage. It can be used as an alternative to excitotoxins such as ibotenic acid and QA. QA is formed by tryptophan metabolism via the kynurenine pathway. Tryptophan shares other neutral amino acids that are used as transporters to cross the blood–brain barrier [27]. Tryptophan is absorbed and converted into kynurenine in the brain by astrocytes, macrophages, microglia, and dendritic cells. The enzyme 3-hydroxy anthranilic acid oxygenase converts kynurenine into QA during the enzymatic process. Normal QA levels do not cause harm. However, even minor increases in QA levels can result in toxicity [28] (Table 1). As kynurenine is converted to QA at toxic levels in HD striatum, this may facilitate neuronal death. Interestingly, QA promotes an increase in the HTT protein in the mouse striatum [26]. It is imperative that QA be applied directly to the striatum since it cannot penetrate the blood–brain barrier. It promotes striatal neurodegeneration in rats by injecting excitotoxin kainic acid into the striatum, which kills MSNs selectively [25]. The selective NMDAR agonist QA was also used to mimic HD symptoms in rats and non-human primates. This was due to its ability to induce selective degeneration and morphological abnormalities in MSNs, including dendritic spine loss. In terms of the mechanism of action, QA produces an increase in Ca^2+^ influx, a decrease in ATP synthesis, and consequent excitotoxic cell death in the NMDA-receptor-rich striatum. QA neurodegeneration resembles key characteristics of neurodegeneration associated with human disease [29]. QA can be used in non-human primates, which could be a more reliable model for HD. The QA model impacts cognitive performance, making this model more like HD than others.

### 3.2. Metabolic Models

3-NP is a toxin that inhibits the mitochondrial enzyme succinate dehydrogenase irreversibly. When animals are treated with this toxin, they develop severe dystonia due to cell death in the caudate and putamen. 3-NP simulates a downstream pathway of cell death identified in the HD brain. The 3-NP model is characterized by mitochondrial dysfunction and is considered reliable for mimicking several aspects of HD (Table 1). Reduced ATP production is the consequence of impaired glucose metabolism in brain cells due to enzyme shortages. In the brains of HD patients, several enzymes involved in the tricarboxylic acid cycle and the electron transport chain are downregulated [26]. The 3-NP model can be obtained by injecting peripherally into rats, mice, and non-human primates as the compound crosses the blood–brain barrier. This is significant for 3-NP dosing because treatment with the compound caused a significant reduction in rat weight (approximately 20 g per day). The effect of 3-NP treatment is different on different rat strains [30].

Depending on the time of treatment administration, the 3-NP model can replicate both the hyperkinetic and hypokinetic HD symptoms. Although 3-NP is taken systemically, rats experience hyperkinetic symptoms like those found in early- to mid-stage HD when given two separate doses for promoting selective degeneration of neurons in the lateral striatum [31]. This model indicates that mitochondrial injury is more prevalent in neurons of the lateral striatum. In HD brains, 3-NP induces both necrosis and apoptosis, resulting in cell death. A wave of necrotic cell death occurs immediately after a 3-NP injection, followed by gradual apoptosis [26].
biomedicines-11-03331-t001_Table 1Table 1Non-genetic rodent models of HD.Animal ModelStrain
Behavioral and Neuronal ChangesReferenceExcitotoxin model: Quinolinic acidRat (Sprague-Dawley and Fischer),Mouse and non-human primateExcitotoxicityHyperkinesia, apomorphine-induced dystonia and dyskinesia, spontaneous dyskinesia with higher doses.Poor memory recallVisual–spatial deficienciesProcedural memory deficits[32,33,34,35,36]Metabolic models: 3-Nitropropionic acidRat (all except Fischer), mouse, non-human primatesMitochondrialimpairment by irreversiblyinhibiting succinate dehydrogenaseHyperkinesia (low dose), hypokinesia (high dose), apomorphine-induced dystonia and dyskinesia, spontaneousdyskinesia with long-termadministration.Defects in ORDT in non-human primates Rats’ working memory and reference memory tested with radial arm water mazesOpen-field apparatus habituation impairments[30,37,38]


## 4. Genetic Rodent Models of HD

### 4.1. N-Terminal Fragment Mouse Models

The first type of transgenic HD rodent model was developed utilizing a 1.9 kb human genomic fragment which was overexpressing exon 1 of the human gene encoding huntingtin (IT15) with long (141–157) CAG-repeat expansions (termed R6 mice) [39]. The human HTT gene contains enlarged CAG repeats and the first 262 bp of intron 1 [40]. In the R6/1 model, there are 115 CAG repeats, while in the R6/2 model, there are 145 CAG repeats [41]. Other N-terminal transgenic lines developed include the R6 [42,43], the N171-82Q, and the Tg100 lines [44]. A wide range of applications is possible with the R6/2 and N171-82Q lines. The R6/1 and R6/2 lines carry one copy of a human genomic segment. The segment contains the promoter sequences of HTT, exon 1 HTT, and approximately 200 bp of intron [45]. There is an expression level of approximately 31% and 75% of natural HTT in the R6/1 and R6/2 lines, respectively. It is translated to produce an exon 1 HTT protein (Table 2).

The R6/2 transgenic model is the most thoroughly researched. Behavioral and motor impairments in this animal model can be evaluated as early as 5–6 weeks of age [39]. Synaptic plasticity deficits at hippocampal CA1 synapses as well as electrophysiological changes in the cortico-striatal circuit [46] have been identified, as well as cell loss in the striatum. There has been evidence that brain weight and volume reduce over time, beginning at postnatal days 30 and 60. Approximately 90 days after birth, striatal neurons had a significant loss and shrinkage. These pathological changes were associated with changes in body weight, rotor performance, grip strength, and dystonia [47].

Compared to yeast artificial chromosomes (YACs), bacterial artificial chromosomes (BACs), and knock-in lines, R6/2 colonies with widely varying CAG tract lengths were the most extensively utilized models due to their early disease onset and rapid disease progression [48]. This model’s behavioral trait is aggressive. Although the average age of beginning symptoms in R6/2 mice is 9 to 11 weeks, some mice have reported signs of symptoms as early as four weeks. The usual death age is 10 to 13 weeks, and animals seldom live for more than 14 weeks [49].

The observation that striatal and cortical neurons, which are primarily damaged by the disease in humans, have an enhanced response to stimulation of the NMDA glutamate receptor in vitro follows the development of cellular aberrations that could potentially cause cell death in R6/2 mice [50]. Several studies have shown that NMDA receptors play an important role in excitotoxicity, promoting striatal neurons’ sensitivity to endogenous glutamate, the neurotransmitter produced by thalamo-striatal pathways and cortico-striatal pathways [51]. A significant reduction in messenger RNA (mRNA) called pre-proenkephalin (an enkephalin marker) and a relevant loss of orexin-positive neurons in the hypothalamus associated with narcolepsy were reported [52]. There are other mice which are genetically derived from the R6/1 line of mice, including a truncated IT15 gene with extended CAG repeats [39], that are akin to R6/2 mice, which have been extensively researched. The shorter CAG-repeat expansion (115 vs. 145) and decreased expression rate of the mutant transgene compared to the wild type are the main differences between R6/1 and R6/2 lines (31% vs. 75%). Staining of NeuN in the striatum did not show loss of neuronal nuclei cells or met-enkephalin immune-positive cells at any age [53]. In the striatum of 5-month-old R6/1 mice, dopamine cAMP-regulated phosphoprotein (DARPP-32) levels were reduced, indicating cellular dysfunction, but striatal shrinkage of dead cells was not observed [54].

N171-82Q is another HD fragment transgenic mouse model developed by inserting the first 171 amino acids from the N-terminal of the human HTT gene into the mouse genome, which expresses a cDNA encoding an N-terminal fragment (171 residues) of huntingtin with 82 glutamine residues under the regulation of a mouse prion promoter [55]. Having a life expectancy of approximately six months, they appear normal at birth but develop tremors, hypokinesia, a lack of coordination, and a failure to gain weight over time. Occurrence of CAG repeat expression and/or length has a significant impact on illness severity. Animals with longer CAG repeat expansions exhibit more relevant neuronal pathology than mice with shorter repeat expansions [39]. In comparison to R6 mice, this model has fewer (82) polyglutamine repeats, resulting in a later onset of symptoms, making it more useful for mimicking adult-onset HD. However, the line 81 mouse model (available commercially from Jackson Laboratories) has a shorter lifespan (5 to 6 months). As long as they are under 2.5 years of age, their behavior is completely normal. A report from week 11 indicates that N171-82Q mice exhibit developmental deficits such as resting tremor, hypokinesia, hindlimb clasping, and abnormal gait [26], symptoms of behavioral syndrome. The N171-82 HD mice were found to have an increased striatal cell volume at 16 weeks of age, and a mutant HTT gene was found to be included in the striatum, cortex, and hippocampus between 16 and 20 weeks of age [56]. Transgenic mice expressing the steady-state level of human HTT protein fragment terminating at residue 586 were generated with 82 CAG repeats to generate the N586-82Q model, which exhibits a robust phenotypic profile and relatively stable phenotypes over generations [57].

As a result, heterozygote mice were developed with normal (CT18) or enlarged (HD46, Tg100) glutamine repeats expressing the N-terminal one-third of huntingtin. Between 3 and 10 months, HD mice developed motor impairments. The onset age was determined by the length of the enlarged polyglutamine. The phenotype severity increases with age. The onset of the phenotype was anticipated by striatal modifications (nuclear inclusions), whereas the onset and severity of behavioral abnormalities were predicted by cortical changes, particularly the accumulation of huntingtin in the nucleus and cytoplasm and the formation of dysmorphic dendrites. Striatal neurons in HD mice display different responses to cortical stimulation and NMDA activation. In Tg100 neurons, NMDA enhanced intracellular Ca2+ levels compared to wild-type neurons [44].

### 4.2. N-Terminal Rat Length Fragment Models

At present, only TgHD rats are used as a transgenic rat fragment model. It is important to note that these animals are transgenic, as they express a human/rat combined fragment of the HTT gene [58]. CAG repeats are relatively few, which results in phenotypes that appear gradually throughout adulthood. A rat endogenous promoter, which is included in the construct, is responsible for the regulation of 51 CAG repeats in the fragment. Mutant fragments are predominantly expressed in the basal ganglia, frontal and temporal cortices, hippocampus, and midbrain, with considerably lower levels in the cerebellum and spinal cord [59] These rats, like other transgenic animals, appear normal at birth but have a weight lower by about 20% than age-matched wild-type rats [60]. TgHD rats have been bred on a Sprague-Dawley lineage, as opposed to commonly used mouse fragment models (e.g., R6/2). TgHD rats exhibit a slow and late-onset disease phenotype [59]. There is a lower level of HTT production in the transgenic TgHD rat fragment than in the endogenous rat fragment. This results in the TgHD rat being considered an effective model for adult-onset HD, which is prevalent among patients. In the cerebellum and spinal cord, the translated protein is relatively low in expression, but can be detected in most parts of the central nervous system. It forms HTT-containing protein aggregates throughout development [60,61,62]. Approximately at the age of six months, this becomes apparent, with the nucleus accumbens damaged. As the animal ages, additional brain regions are affected. Aggregation occurs largely in the caudate-dorsomedial putamen portions at approximately nine months of life [59,60] (Table 2). The prenatal period has been associated with enlarged ventricles and reduced striatal volume in some studies [63], but these phenotypes have not been detected in others [64], even in 18-month-old rats [65]. There is evidence that degenerated neurons are present both in the cortex and the striatum, as well as a reduction in the number of striatal neurons at the age of 12 months [60,61]. A variety of neuropathological characteristics, such as striatal neuron shrinkage, have also been documented.

In various behavioral tests, TgHD rats were found to be less stressed than wild-type rats, and there were also signs that rats had poor spatial working memory as early as six months of age. There has been evidence of increased mortality in BACHD rats at 24 months of age [59,60]. BACHD rats are a generic manipulation of the transgenic Human HTT (97 CAG/CAA) full-length human protein. Although these rats are obese, their stability and body weight remain unchanged. The presence of progressive brain atrophy is commonly associated with smaller brains, although the interaction between progressive atrophy and developmental deficits remains unclear [66]. Most of the CNS is characterized by aggregate formation and morphological changes, including deteriorating neurons. About premature death, nothing was reported until 16 months, and the motor phenotypes were impaired rotor performance and gait abnormalities. A reduction in exploration anxiety was associated with psychiatric conditions. Cognitive disorders have been observed to affect performance in several areas, although most of the results are preliminary and have not been extensively reproduced. The development of aggregates can be observed around three months of age following the onset of phenotypes. Anxiolytic behavior can be detected by four months of age, while impairments on the rotarod can be detected after one month of age [67].

### 4.3. Full Length Mouse Models

Full-length transgenic models carry the full-length HTT sequence and express full-length HTT protein containing expanded polyglutamine repeats. By introducing mutant genes from humans into yeast artificial chromosomes (YACs) and bacterial artificial chromosomes (BACs), transgenic mouse models that express mutant versions of the full-length HTT protein have been generated [68]. YAC generated the first Huntington full-length mouse model with 18 glutamines [18], and YAC46 and YAC72 mice that contain 46 and 72 CAG repeats, respectively [69]. In the YAC128 transgenic mice, human HTT is duplicated four times, with 125 CAG repeats interrupted by CAACAACAACAGCAA at positions 24 and 28 and 109 and 113, respectively. The YAC128 model with 128 glutamines has been developed in order to improve expression of the phenotype [68]. This mice phenotype with a higher number of glutamines develops symptoms mimicking those of HD earlier and more pronounced when compared to the first YAC models.

Two other models are represented by the FVB/N and C57BL/6 mice, with the FVB/N line being the most affected [70]. From the age of 3 months, they acquire motor abnormalities, cognitive impairments, striatal atrophy, and neuronal loss [71] (Table 2). Two BAC models of HD have recently been developed. Human HTT is expressed in the BACHD model [72] with 97 CAG/CAA repeats, while the BAC-225Q model expresses human HTT with 97 CAG/CAA repeats. A newly developed model contains the full-length mouse Htt with 225 CAG repeats [73] and is driven by the mouse Htt promoter. It will be discussed later how the YAC128 and BACHD models develop motor abnormalities as well as cognitive and neuropsychiatric disorders. BACHD and YAC128 models exhibit gradual motor degeneration [48,71,73]. They have a shorter rotarod fall latency and develop balance and gait problems later in life [48,68,71,74]. At the age of 3 months, YAC128 mice are hyperactive, whereas at 12 months of age, they are hypoactive [68], and motor deficits begin before the hypoactive phase. Young BACHD mice do not exhibit the early hyperactive phenotype, which is seen in YAC128 mice, where it shows decreased activity in the open field from 6 months onwards [75]. It is possible that some neuronal dysfunction may already be evident at the outset of motor impairment because motor performance at 6 months is correlated with neuronal loss a few months later. Dysfunctions in cortical and hippocampal neural networks precede motor symptoms in HD [76,77]. There are several learning deficits in BACHD and YAC128 mice, including motor learning, novel object recognition memory [78,79], sensorimotor gating [80,81], reversal learning, and strategy shift [82,83,84,85]. The generation and characterization of a novel HD mouse model BAC226Q by using a bacterial artificial chromosome (BAC) system has been reported by Shenoy SA et al. (2022) [86]. BAC226Q mice gradually showed HD-like psychiatric and cognitive phenotypes at 2 months. BAC226Q mice presented motor deficits from 3 to 4 months. At 11 months, BAC226Q mice presented neuropathology like brain atrophy, particularly in the cortex and striatum, striatal neuronal death, extensive huntingtin inclusions, and reduced life span [86].

### 4.4. Full-Length Rat Models

An HD transgenic rat model was created using human BAC containing the full-length HTT genomic sequence with 97 CAG/CAA repeats and all regulatory regions. BACHD transgenic rats showed a relevant, early-onset, and progressive HD-like phenotype, including motor impairments and anxiety-related symptoms. BACHD rats did not gain weight compared to BAC and yeast artificial chromosome HD mice models expressing full-length mutant huntingtin. Several studies observed that, neuropathologically, the distribution of neuropil and nuclear accumulation of N terminal mutant huntingtin in BACHD rats is identical to what is observed in the human HD brain [66]. More recent studies focused on hemizygote rats with Sprague-Dawley backgrounds [87]. Initially, two distinct lines (TG5 and TG9) were developed, and each over-expressed the transgene to varying degrees. It has also been found that reduced TFIID creation may play a role in the deregulation of gene expression in the striatum of both transgenic lines TG5 and TG9 in BACHD rats as early as three months [88]. At 13 months of age, there is heavy damage to areas such as the cerebral cortex, the hippocampus, the amygdala, and the nucleus accumbens. According to studies conducted on the TG5 lines of BACHD rats, the striatum shows a relatively lagging phenotype compared to the cerebellum as early as 2 months of age [66,89]. Reversal learning abnormalities in BACHD rats have been discovered [90]. At 3 months of age, BACHD rats displayed some impulsive behavior [87,91]. Food consumption is lower in BACHD rats than in wild-type rats. Male BACHD rats have been observed to be obese without increasing body weight [33] when their metabolic features are considered. Although mortality beyond 17–18 months has not been assessed, a possible shorter lifespan for BACHD rats has not been studied yet [66,87].
biomedicines-11-03331-t002_Table 2Table 2Transgenic animal models of HD.Transgenic Model
Animal ModelStrainCAG Repeat LengthLife SpanBehavioral and NeuronalAlterationsReference**N-terminal****fragment length mouse model**R6/2C57BL/6J116Reduced life span(10–13 weeks)Progressive decline from 5 weeks rotarod (5–6 weeks). In the open field, anxiety-like behavior (8 weeks). Learning and memory deficits worsen over time. Sleep disturbances (9 weeks). Significant reduction in neurons and neuronal atrophy in the striatum within 3 months.At 4 weeks, neurons had intranuclear inclusions in the cortex.Reduced brain volume.[43,44,47,92,93,94]DBA/2J116B6CBA/CaMixed128B6CBA/CaMixed160C57BL/6J168DBA/2J168C57BL/6J251DBA/2J251C57BL/6J293DBA/2J293R6/1C57BL/6J,BALB/cByJ,B6CBA/Camixed116Reduced life span(32–40 weeks)Progressive decline rotarod (8 weeks). Reduced activity open field (23 weeks) Anxiety-like behavior open field. Spatial learning deficits in Morris water maze. Sleep disturbance.Decrease in striatal volume. Presence of cellular inclusions at 5 months of age. Neuronal intranuclear inclusions.Neurotransmitter receptor level changes.[42,54,95]N171-82QC57BL/6×B6C3H/He J mixed82Reduced life span(Line77—2.5 months, lines 81 and 100—5–6 months, Line 6—8–11 months)Progressive decline, gait abnormalities, hypoactivity, depressive-like behavior, and working and reference memory deficits at 14 weeks of age.Shrinkage and neuronal loss in the striatum, hippocampus, and frontal cortex at 4 months of age.Neuronal intranuclear inclusions. Reduced brain volume. Dopamine and serotonin levels remain unchanged.[47,57,68]N586-82QC57BL/6JB6C3H82Reduced life span (8–9 months for N586-82Q-63C)Progressive decline in rotarod, and open field consecutively from 3 and 4 months. Asymmetric dyskinesia (4 months onwards).Fear conditioning causes contextual and cue-dependent memory deficits at 8 months.Large inclusions in every part of the brain, astrogliosis in cerebellum, striatum, and cortex. Granule cells in the cerebellum are gradually degenerating. Total brain capacity is reduced. Cerebellar and hippocampal atrophy occurs.[55,57,96]Tg100B6SJL mixed100Reduced lifeExpectancy(10–12 month)Cortical changes, formation of dysmorphic dendrites, enhanced intracellular Ca^2+^ levels.[59,60,61]**N-terminal****fragment length rat model**TgHDSprague-Dawley rats51Increased mortality at 24 months of age.Around the age of 6 months, the development of aggregate becomes visible. Anxiolytic behavior can emerge as early as 2 months of age, while motor and cognitive phenotypes typically appear around 6–9 months.In specific brain regions, structural changes, neuronal loss, and atrophy.Impaired spatial working. Reference memory, attentional deficiencies[97]BACHDSprague-Dawley rats97Reduced lifeExpectancy(16–20 weeks)Alterations in structure and the presence of decaying neurons, gait anomalies, and rotarod performance. Reduced exploration anxiety. At around 3 months of age, visible aggregate development begins. The onset of behavioral phenotypes differs, with rotarod deficits visible at 1 month and anxiolytic behavior visible at 4 months.Frequently reported smaller brain volume.[72,87,89]**Full length****mouse model**YAC128C57BL/6FVB/N128Reduced lifeexpectancy in maleDeficits on T-maze. At 3 months of age on an open-field test, they display hyperkinesia, a 50% reduction in body weight when compared to wild-type littermates. Circling behavior, gait problems, ataxia, and hindlimb clasping are all common symptoms. The rotarod test shows a steady deterioration. Hyperactivity (3 months) followed by hypoactivity (12 months). Depressive-like phenotype (3 months) and sucrose consumption.Cell loss ranges from 18% to 40% primarily in the lateral striatum.Increased HTT staining in the nucleus.The striatum, nucleus accumbens, cortex, and cerebellum neurons show intranuclear inclusions (15 months).[68,70]BAC97YAC18(Hu97/18)FVB/N18, 97YAC128, BAC21(Hu128/21)FVB/N125BACHDFVB/NJ97Normal life spanDecreased motor learning over time, the rotarod (2 months): progressive deterioration Catwalk examination of gait impairments (9–10 months).Object recognition memory impaired (6 months).Deficits in reversal learning and strategy shift water (9–10 months) T maze and cross maze. Startle response is disrupted by sensorimotor gating abnormalities (9 months), inhibition of the prepulse (7 months). Decrease in activity in the open field (from 6 months).Inclusions of mHtt in the brain and striatum (12 months) There is no neuronal loss. A reduction in cortical and striatal volume (12 months).[68,71]BAC22Qmouse model-Normal life spanBAC226Q mice gradually showed HD-like psychiatric and cognitive phenotypes at 2 months.BAC226Q mice presented motor deficits from 3 to 4 months.At 11 months, BAC226Q mice presented neuropathology like brain atrophy, particularly in the cortex and striatum.[86]**Full length****rat model**BACHDSprague-Dawley rats97Normal life spanTransgenic rats perform better on the rotarod test than wild-type littermates, but their performance deteriorates over time. At 10–15 months: head dyskinesias and gait abnormalities. 12 months: Traveling over an elevated beam takes longer. Compared to wild-type rats, at the age of 24 months, lost 20% of their weight. Deficiencies on the radial arm and high plus mazes in the age of 12 months.At 8 months of age, the lateral ventricles are enlarged. At 12 months: Inclusions in the brain and 2 months.[60,87,89]


## 5. Knock-In Mouse Models

It is noted that knock-in (KI) mouse models are more genetically precise than transgenic mouse models, since they carry one or two copies of the mHTT gene. The characteristics of these mice are like those of HD, and they have a late-onset phenotype, which makes them suitable for investigating early neurophysiological changes that contribute to behavioral alterations [98] (Table 3). Until now, two different methodologies have been used to develop KI models. The mouse polyQ encoding sequence (CAG)2CAA(CAG)4 was replaced with an expanded repeat of approximately 150 CAGs in HDhQ150 mice (also known as CHL2) [99]; HDhQ200 [100] and HDhQ250 [101] were the first to be developed. This model contains 50-365 CAG repeats derived by selective breeding, which have been better characterized and described here. As a result, the major variation between the targeted and normal alleles was the CAG repeat tract. These mice can be bred to homozygosity or used as heterozygotes. Congenic lines of C57BL/6 and CBA were also developed [92]. As a result of inserting a chimeric HDh gene homolog (HDh)/human mHtt exon one into the exogenous murine HDh locus, which was controlled by the mouse Htt promoter, the HDhQ20, HDhQ50, HDhQ92, and HDhQ111 KI lines were developed [102]. The CAG140 [103] and zQ175 models have been developed using the same technique, with the latter resulting from spontaneous growth of the CAG repeat number in the CAG140 model [104]. ZQ175 has a shorter lifespan than HDhQ150, HDhQ111, and CAG140, while they have a normal lifespan [105]. A series of electrophysiological studies conducted on homozygous mice at 3 months of age and heterozygous mice at 4 months revealed gradual hyperexcitability in medium spiny neurons in ZQ175 animals, including reduced glutamatergic transmission and decreased striatal and cortical volumes [106]. On gametic, the CAG repeat shows inconsistent transmission, particularly in the context of the C57BL/6 background. This background has CAG repetition lengths of 50, 100, 150, 200, 250, 315, and 365. In HD, the HDhQ150 model recapitulates several disease-related features evident in early pathogenic alterations. Cognitive deficits occur after 24 weeks [107]. At four months of age, HDhQ150 mice were found to have diminished grip strength and weight loss was observed at 12 months, and rotarod motor deficits were observed at 18 months [92,108]. Approximately five months after birth, HDhQ150 mice begin to display signs of aggregation in the striatum and olfactory tubercle [109]. KI mouse models exhibit varying degrees of motor impairment, decreased activity, rotarod impairment, and gait abnormalities evident with age. Hdh150 has been found in homozygous mice before they develop serious motor problems related to cognitive deficits, which often appear before motor deficits [105]. KI models HDhQ111, CAG140, and HDhQ250 have been reported as neuropsychiatric phenotypes. The HDhQ111 mouse exhibits different types of alterations around the age of 3–4 months in both males and females [110]. At four months, CAG140 exhibits impairments in long-term recognition memory, which could be interpreted as an early non-permanent anxious-like phenotype similar to depressive behavior. Contrary to HDhQ250, HDhQ250 displays a non-progressive depressive phenotype similar to that of HD patients [94,111]. The KI models have a range of HD-specific neuropathological traits, with each model displaying unique abnormalities similar to human HD. HDhQ111 did not have striatal atrophy, but it does have nuclear localization of mHTT, particularly in medium spiny neurons, as well as the production of N-terminal inclusions and insoluble aggregates [112]. It has been observed that both nuclear and neuropil aggregates are present in the CAG140 model, starting from the dorsal striatum, nucleus accumbens, and olfactory tubercle. When CAG140 mice are 12 months old, they show reduced levels of DARPP32 in the striatum and gliosis in the cortex [103]. Gliosis has also been detected in the striatum by 23 months, and this is evident in striatal atrophy, loss of mature and immature neuronal spines, and reduction in dendritic complexity [111]. The ZQ175 brain exhibits striatal atrophy, mHTT inclusions, decreased striatal dopamine and brain-derived neurotrophic factor (BDNF) levels, and cortical thinning [113].

## 6. The Need for Large Animal Models of Huntington’s Disease

Rodent models of HD described above provided valuable insight into the common pathological hallmark of misfolded protein aggregates or inclusions. Due to their short lifespan, they are not suitable for the study of HD evolution. Conversely, many genetic mouse models do not exhibit the striking neurodegeneration associated with HD [117]. Since small and large animals differ significantly in terms of species, large animal models of HD can assist in identifying the similarities and differences between human disease, and animal models and can therefore facilitate the identification of therapeutic targets that are more effective. However, creating genetic models for huge animals is more difficult than developing rodent models. Embryonic stem cells (ESCs) from pigs, monkeys, or other large mammals have yet to be discovered that can be utilized to create gene-targeted animals. Induced pluripotent stem cells (iPSCs) are similar to embryonic stem cells (ESCs) and are capable of being used to manipulate endogenous genes and to target specific genes in animals of various types (Table 2). In contrast, the use of iPS cells for gene targeting is still in its early stages of development [118,119,120]. Currently, most transgenic monkey research is related to the infection of fertilized oocytes with lentiviral vectors and the transplant of embryos. This necessitates many donors and surrogate monkeys. Nuclear transfer, a cloning process with a poor percentage (1–2%) of transferred pig embryos developing to term, can also be successfully used to generate transgenic pigs [121]. Furthermore, the expense of maintaining and producing large animals, as well as ethical considerations and tight regulations around their usage in scientific research, make their utilization problematic. Gene targeting in large animals has opened new possibilities thanks to recent gene targeting developments. TALEN and Cas9, two new techniques developed to alter gene expression, use transcription activator-like effector nucleases (TALENs) and Cas9 endonucleases from the type II bacterial CRISPR/Cas system, respectively. It is possible to modify genes in embryos without ESC by using TALENs or Cas9 [122]. This novel approach employs DNA-binding peptides that attach to specific target DNA sequences, allowing nucleases to cleave the DNA, resulting in gene loss [123]. Because of the difficulty and cost of generating and characterizing large animals, rodent models of neurodegenerative diseases will continue to be a major modeling system for investigating a variety of diseases. The use of large animal models, however, has the potential to provide a more rigorous mechanism of validating the relevance of critical findings in small animal models. Rather than relying on overexpression of N-terminal mutant HTT for confirmation, a knock-in technique would need to be employed to express endogenous N-terminal mutant HTT. Transgenic models of key neurodegenerative illnesses using higher mammalian species or large animals would provide a further understanding of the pathophysiology of these diseases. Furthermore, given the high rate of failures in clinical trials of compounds effective in small animal models, big transgenic animals could provide a more reliable mechanism for confirming therapeutic efficacy before starting human trials.

## 7. Conclusions

High-resolution modeling in the mouse has proven successful. The substantial CAG repeats expansions used in these models might trigger early illness onset in humans [124]. Therefore, because they mimic childhood-onset diseases rather than late-onset diseases, mice tend to display robust phenotypes throughout their lifetime [125]. The most extensively utilized rodent models for studying HD pathogenesis and potential treatments are the rodent models summarized in Figure 2. Neurotoxins were used to develop non-genetic models. Transgenic mice that express the mutant HTT protein or knock-in mice that replace the mouse gene or a portion of it with a mutated gene were used to develop genetic rodent models of HD [110]. These are considered “gold-standard HD models”. However, no contemporary model can claim to be the leading model due to the presence of overt front striatal cognitive deficits and the lack of significant striatal degeneration [60]. The purpose of the study should determine the duration, sample size, and extent of linkage to human HD at the biochemical and behavioral levels. Another issue is the consistency and stability of genetic variation as a result of breeding and background strain. These issues are not only relevant for current use but also for the development of new animal models for HD and other neurodegenerative disorders in the future.

The choice of an animal model depends on the specific aspect of the disease investigated. Toxin-based models can still be useful, but most experimental hypotheses, especially those involving therapeutic interventions, necessitate success in a genetic model, whose choice is determined by the experimental question. As described above, there are animal models showing several similarities with HD symptoms or pathologies. Chemically induced HDs, genetic HDs involving cell-free and cell culture, lower organisms (such as yeast, *Drosophila*, *C. elegans*, zebrafish), rodents (mice, rats), and non-human primates are all included in this group. In addition, these models provide an accessible means of studying molecular pathogenesis and testing potential treatments. Among these models, 3-NP-induced and R6/2 transgenic mouse models are most commonly used. As HD is an inherited disorder caused by gene mutation, genetic models are closer to the human condition than chemical lesion models. Compared to other genetic models, KI mice have the advantage of carrying the mutation in the appropriate protein context—the full-length Huntington protein—and under the endogenous promoter, which makes KI mice the most valuable models at present. However, the development of more effective treatments may not come along until better animal models are available and provided for evaluation of a drug’s efficacy. Genetic models are invaluable, and the seminal multidisciplinary work promises a promising future to understand HD pathological mechanisms and create more effective treatment opportunities. Despite this promising beginning, none of the models replicate the massive cell loss of striatal neurons that occurs in patients. There is still a long way to go to create mechanistically significant, full models of this disease. A combination of novel genetic approaches to obtain a clearer picture of the environment where the disease develops in humans is required. Since HD outcomes come from a mutation in one gene and that mutation can differ between patients, differences in disease progression have been established. The difficulty in finding effective treatments for HD may also depend on the absence of a suitable laboratory model that recapitulates the complexity of the disease. In view of the above limits, the use of stem cells to mimic the conditions of different forms of HD and to use them to screen different drugs for future testing in patients may represent a promising direction of future research. Novel approaches can be found in the use of iPSC from HD patients. Genetically modified iPSC may produce the correct Huntington protein and could be used as a personalized treatment for single HD patients.

## Figures and Tables

**Figure 1 biomedicines-11-03331-f001:**
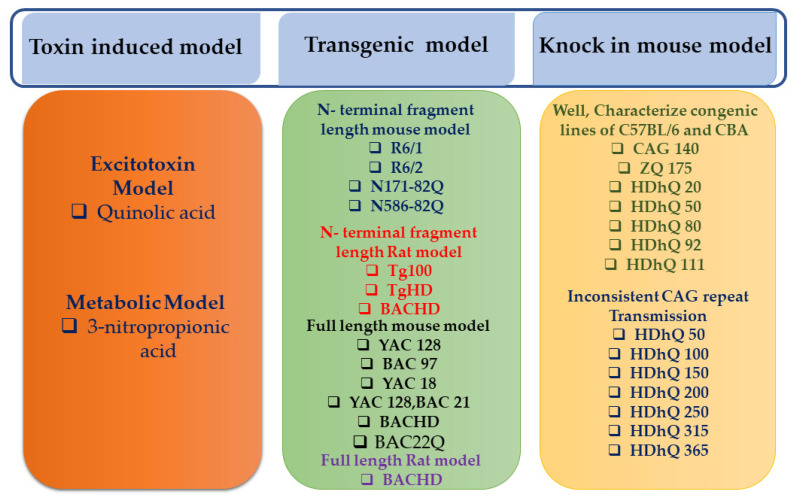
Schematic classification of the animal models developed for modeling HD. The colors highlight the different type of animal model analyzed in the review.

**Figure 2 biomedicines-11-03331-f002:**
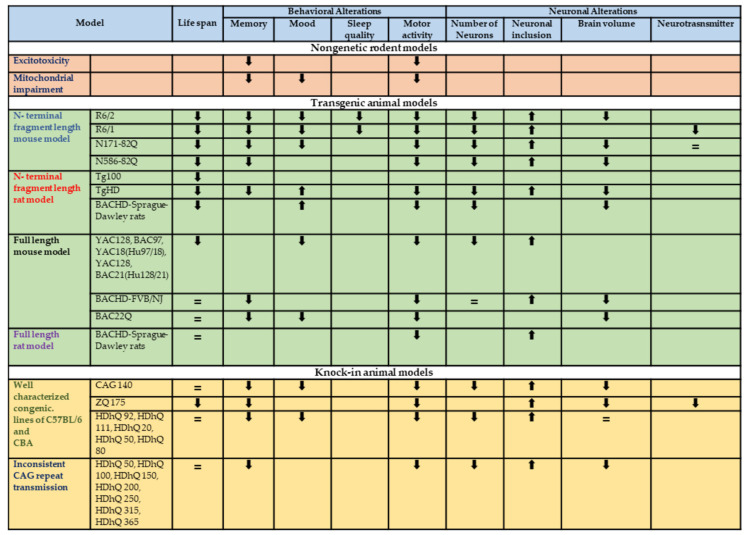
Schematic representation of the main characteristics of the different HD animal models. The arrows indicate the changes in the behavioral and neurological alterations compared to the control animals. =: no modifications in the parameters compared to the control animals. The blank box indicates a parameter not evaluated in the studies examined. The colors highlight the different type of animal models analysed.

**Table 3 biomedicines-11-03331-t003:** Knock-in animal models of HD.

Knock-In Mouse Model						
	Animal Model	Strain	CAG Repeat Length	Life Span	Behavioral and Neuronal Alterations	References
**Well characterized congenic.** **lines of C57BL/6 and** **CBA**	CAG 140	C57BL/6J	140	Normal life span	Hyperactivity in 1st month and hypoactivity at 4 months. Vertical pole, non-accelerating rotarod, running wheels—sensorimotor performance deficit (4 months), irregularities in gait (12 months). Anxiety-like phenotype, no depressive behavior, long-term memory recognition impairment. Nuclear and neuropil inclusions in the striatum, cortex, hippocampus, and cerebellum at the age of 2 months. Loss of neurons (23 months). At 20–26 months, the corpus callosum volume is reduced.	[48,114]
	ZQ 175	C57BL/6J	198	Reduced life Expectancy in (19 months)	Progressive reduction in open field (2 months), grip strength (1 month homozygous), phenotype (4 months homozygous), climbing activity (8 months homozygous), cylinder test (1 month heterozygous), nesting (heterozygous, 16 months). Two options for procedural learning loss Test swimming ability (homozygous, 10 months). Impairment of executive function test (7 months). Two-choice visual discrimination test for cognitive flexibility (7 months). Nuclear inclusions with a slowed circadian cycle (heterozygous). Atrophy of the striatum. Cortical thinning is a condition in which the cortex thins out. Reduced amounts of dopamine and BDNF	[104,106]
	HDhQ 92	C57BL/6J	92	Normal life span	Testing for depressive-like phenotype in women with splash tests and forced swim tests. Open field test for anxiety-like phenotypes (male). Deficit in olfactory function due to an anxio-depressive-like phenotype. Discrimination in the workplace (males). The rotarod of motor learning has been altered. Memory impairment for long-term object recognition (4 months). Impairment of spatial memory (8 months). Working memory and reversal learning deficits non-matching to position tasks and delayed matching to position tasks (8 months). There is no evidence of striatal degeneration. At 4.5 months, Huntingtin protein translocation to nucleus and appears punctate. Striatal gliosis and gait problems at 24 months of age, no overt symptoms. There were no alterations in spontaneous locomotion that could be detected. Inclusions in mHtt (12 months). Loss of neurons (24 months). There is no striatal atrophy.	[115]
	HDhQ 111		111			
	HDhQ 20		20			
	HDhQ 50		50			
	HDhQ 80		80			
**Inconsistent** **CAG repeat** **transmission**	HDhQ 50	C57BL/6J	50	Normal life span	Learning problems in the spatial and reverse directions in the water maze with three stages (4 and 8 months, respectively). Performance issues with extra-dimensional shifts (6 months). Reduced sensitivity to startling stimuli (6 months homozygous). Balance beam performance is impaired (homozygous for 4 months, heterozygous for 12.5 months). Grip strength test with decreased muscular strength (20 months), irregularities in gait (15 months). On the rotarod, no impairment was seen. Water mazes with impaired spatial and reversal learning (4 and 8 months, respectively). Impaired performance in extra-dimensional shifts (6 months). Reduced sensitivity to startling stimuli (6 months homozygous). Intranuclear inclusions are a type of intranuclear inclusion (13 months). Gliosis (14 months, heterozygous). Loss of neurons (12.5 months). Atrophy of the striatum (12.5 months). Reduced D1 and D2 receptor binding (only homozygous). Two types of striatal and cortical astrogliosis (20 months). Abnormalities of the cerebellum Purkinje neurons.	[99,111,116]
	HDhQ 100		100			
	HDhQ 150		150			
	HDhQ 200		200			
	HDhQ 250		250			
	HDhQ 315		315			
	HDhQ 365		365			

## Data Availability

Not applicable.

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
