# Peer review of "Rodent Models of Huntington’s Disease: An Overview"

_biomedicines, 2023, doi:10.3390/biomedicines11123331_

Round 1

Reviewer 1 Report

Comments and Suggestions for Authors

This review manuscript is one of many recently published in the field of rodent HD mouse models. Although this review manuscript is well written and summarizes current status of HD mouse models, it has to be noted that this manuscript is additional replication of many review articles published in recent years, discussing HD rodent models pros and cons. It will be more attractive to the readers if authors present current knowledge of rodent HD models in more synthetic way e.g. by showing clear differences between models as a pictogram instead of tables. 

Comments on the Quality of English Language

There are number of gramma and spelling mistakes that should be corrected. 

Author Response

See enclosure

Reviewer 2 Report

Comments and Suggestions for Authors

The authors examined in detail models that can partially reproduce Huntington's disease in various animal species. In conclusion, the authors discuss in detail the problems of individual models and conclude that none of the modern models reproduces massive cell loss of striatal neurons.

In my opinion, the authors conducted a very good analysis and clearly structured the material.

There are a few small comments about the work:

1. The abstract turned out to be very long and is mainly devoted to the description of the disease itself. In my opinion, in the abstract it is necessary to devote part of the text to the main conclusions that the authors made at the end of the article. Because it is not clear from the abstract what the main idea of this scientific article is.

2. Point “4.3 Full length mouse model”, in my opinion, models must be written in the plural, because This section discusses several models.

3. Table 2 – last row - Full length rat model, but this item applies to Sprague-Dawley rats and mouse model (BAC22Q). Perhaps authors need to separate these points.

Author Response

See enclosure

Round 2

Reviewer 1 Report

Comments and Suggestions for Authors

Thank you for partially addressing my suggestions.

Comments on the Quality of English Language

 There are still some gramma issues.